# Surface Acoustic Waves (SAW) Sensors: Tone-Burst Sensing for Lab-on-a-Chip Devices

**DOI:** 10.3390/s24020644

**Published:** 2024-01-19

**Authors:** Debdyuti Mandal, Tally Bovender, Robert D. Geil, Sourav Banerjee

**Affiliations:** 1Integrated Material Assessment and Predictive Simulation (iMAPS) Laboratory, Department of Mechanical Engineering, University of South Carolina, Columbia, SC 29208, USA; dmandal@email.sc.edu (D.M.); tallyb@email.sc.edu (T.B.); 2Department of Chemistry, University of North Carolina at Chapel Hill, Chapel Hill, NC 27599, USA; bob.geil@unc.edu

**Keywords:** surface acoustic waves, lab-on-a-chip sensors, SAW devices, shear horizontal waves sensing, tone burst signal, guided waves, piezoelectric, MEMS, predictive simulation, actuation and sensing, tone burst-interdigitated electrodes

## Abstract

The article presents the design concept of a surface acoustic wave (SAW)-based lab-on-a-chip sensor with multifrequency and multidirectional sensitivity. The conventional SAW sensors use delay lines that suffer from multiple signal losses such as insertion, reflection, transmission losses, etc. Most delay lines are designed to transmit and receive continuous signal at a fixed frequency. Thus, the delay lines are limited to only a few features, like frequency shift and change in wave velocity, during the signal analysis. These facts lead to limited sensitivity and a lack of opportunity to utilize the multi-directional variability of the sensing platform at different frequencies. Motivated by these facts, a guided wave sensing platform that utilizes simultaneous *tone burst-based excitation* in multiple directions is proposed in this article. The design incorporates a five-count tone burst signal for the omnidirectional actuation. This helps the acquisition of sensitive long part of the coda wave (CW) signals from multiple directions, which is hypothesized to enhance sensitivity through improved signal analysis. In this article, the design methodology and implementation of unique tone burst interdigitated electrodes (TB-IDT) are presented. Sensing using TB-IDT enables accessing multiple frequencies simultaneously. This results in a wider frequency spectrum and allows better scope for the detection of different target analytes. The novel design process utilized guided wave analysis of the substrate, and selective directional focused interdigitated electrodes (F-IDT) were implemented. The article demonstrates computational simulation along with experimental results with validation of multifrequency and multidirectional sensing capability.

## 1. Introduction

### 1.1. Sensors and Actuators

Sensors and actuators are the fundamental components of our digital world, especially applicable in automation, biomedical systems, microelectromechanical systems, robotics, and control systems [1,2,3,4]. Sensors and actuators play a crucial role in perceiving the world, acquiring data from the physical world, and then reacting to or analyzing the effects based on it. The sensors are basically, like the “eyes and ears” of the system, and the actuators are the “muscles”. The sensor usually perceives the physical energy in the form of light, heat, sound, or vibration, motion, etc., and converts the analog signals into a potential digital signal for quantification [5]. Both sensors and actuators have made a significant impact on modern technology over the years. They are widely used in different fields. Automation and controls, health care monitoring, environmental monitoring, agriculture, industry, robotics, and security [6,7,8,9,10] are significantly impacted by their application. Although there are different types of commercially available sensors, a few sensors and actuators that are popular in microelectromechanical systems involve biosensing, chemical sensing, gas sensing, microfluidics, RF sensing, etc. [11,12,13,14,15,16]. The sensors mentioned above use different types of sensing and actuation mechanisms, among which the surface acoustic-based technique and its devices are popular in terms of robustness, small size, portability, reproducibility, high-throughput, low power, sensitivity, and selectivity [17,18,19,20].

### 1.2. Biosensors and Types

Biosensors have become potent tools that connect the fields of biology and electronics [21,22,23]. Biosensing is an analytical tool that consists of the biologically active material used in close concurrence with a device that measures and converts a biochemical reaction signal into a quantifiable electrical output proportional to its concentration [24]. These sensors are essential for a variety of applications, including food safety, environmental monitoring, healthcare, and medication development [25,26,27,28]. Biosensors are highly specialized tools that can quickly and precisely identify biological substances and convert them into meaningful signals. The biosensor usually consists of three components: bioreceptors, transducers, and signal processing with a display output [29]. Bioreceptors are the components that are primarily responsible for the specific detection of a target biomolecule. Bioreceptors are usually enzymes, antibodies, molecular-imprinted polymers, etc. The bioreceptors form a bio-complex with the target analytes, which can be antigens, DNA, proteins, spores, etc. [29]. The transducer is the component that usually converts the bio-complex reaction into a measurable signal. Transducers are broadly classified based on electrochemical, magnetic, optical, piezoelectrical, and thermometric mechanisms. Following the transducer, the signal is processed using a processor that involves electronics, which usually incorporates signal amplification, conditioning, analog-to-digital conversion, and further by displaying the results. This allows the quantitative analysis of the concentration of the analytes [30,31].

### 1.3. Piezoelectrical Biosensors and SAW Devices

Piezoelectrical biosensors have attracted attention in recent years. The piezoelectric biosensors are mostly divided into two categories: (1) resonating oscillator-based and (2) surface acoustic waves-based (SAW) [18,32]. Although optical, magnetic, thermometric, and a few electrochemical-based biosensing techniques are effective and widely used, they still face challenges. A few of these technologies are very promising, but are expensive and require complicated setups [33]. Many of these techniques require a laboratory and professional handlers for the diagnostics. Certain techniques, like the PCR method, require the transportation of samples and take about 1–2 days for the diagnostics. This leads to the consumption of time and energy and drives costs. Additionally, a few of these technologies lead to false-negatives and false-positives due to poor sensitivity and selectivity, which are highly undesirable [33]. Considering these issues, SAW-based sensing has made a significant impact, especially in the field of biosensing. SAW devices are robust, miniaturized, and utilize the concept of acoustic wave interaction corresponding to mass loading. SAW devices are very sensitive, even to the slightest change in mass loading, making them a powerful tool for biosensing applications [18].

### 1.4. Conventional SAW Device Challenges and Proposed Tone Burst-Based Sensing Platform

An article [18] on SAW devices primarily discusses the different acoustic waves, their physics, and materials for wave generation. The article also discusses different configurations of the interdigitated electrodes (IDTs) for generating unique surface waves. It also summarizes different applications of the SAW sensors in various fields to be used as sensors and/or actuators [18]. Based on those facts and concepts, in this article, we demonstrate a novel tone burst-interdigitated electrode (TB-IDT) and SH-waves-based sensing platform that has access to multiple frequencies and multidirectional sensing within the same sensing platform. The SAW-grade sensor utilizes the concept of piezoelectricity and the converse piezoelectricity mechanism. In this mechanism, an actuator-interdigitated electrode is excited using an AC voltage, and due to the direct piezoelectric effect, the SAW is generated. These acoustic waves transmit and propagate through the sensing test site on which the functionalization of the target elements (such as biological, chemical, gas, mechanical, etc.) takes place. The wave energy propagates in the substrate and interacts with the other pair of interdigitated electrodes. Due to the converse piezoelectric mechanism, it generates an output voltage [18]. This is the primary concept of SAW-based actuation and sensing. Over the years, the delay line configuration has been the most common and conventional type of SAW-based configuration [18]. Although it is very widely used, it notably suffers from a few challenges. The conventional delay line propagates and senses along one axis (uniaxial). The actuator IDT generates acoustic wavefronts in both forward and backward directions, due to which it suffers half-energy losses. Additionally, it is reported that the delay line configuration endures signal losses like reflection loss, transmission loss, and insertion loss [34,35,36,37], to name a few. Moreover, the quality factor of the delay line configuration is reported to be low. Due to this fact, researchers are working on developing new electrode configurations and resonators for better sensitivity and efficacy [18]. One of the major limitations of conventional delay line sensing is the unidirectional approach of the waves. Due to this fact, the waves in a SAW substrate propagate along an intentional direction, and the sensory IDT must be placed along the same axis of propagation for better sensing [18,34,35,36,37]. As a result, all the delay line sensors are tuned to the natural resonating direction, while the other possible sensitive directions remain unexplored. There are a few scenarios where an unknown or undesired mass loading (analyte) interacts with the wave modes to result in a frequency shift where the desired mass loading interacts with the wave modes in such a way that the desired shift in the central frequency is undetectable. These scenarios lead to false-positive and false-negative sensing [33], respectively.

Most conventional SAW-grade sensors utilize a continuous signal for detection or monitoring in which the phase shift (in degrees) is used for the diagnostics. Conventional SAW sensors use a continuous signal in which only the shift in the central frequency peak is considered [18,38] for effective diagnostics. The parameters are effective, but they narrow the boundaries for effective, sensitive sensing. SAW-based biosensors use the concept of acoustic waves on top of a piezoelectric wafer, which is essentially anisotropic [39]. As mentioned earlier, there are cases where corresponding to a particular actuation frequency, mass loading, direction of wave propagation, geometry, and other factors may result in no shifts in the central frequency [33]. In such cases, although the target analyte is present, the sensor displays negative results, thus leading to false-negative scenarios. The situation can be reversed, leading to false-positives, which are also highly undesirable. These result in poor sensitivity and selectivity.

To circumvent these issues, in addition to the central frequency, there is an absolute need to access additional frequencies. There are ample chances that multiple frequencies correspond to different resonances with respect to a certain concentration (mass loading) of a target analyte in a solution. In other words, the chances of a target analyte escaping from a broader range of frequency spectrum-based detection with varied amplitudes of signal responses are very low, thus incorporating into a highly sensitive platform. Here, a 36°YX-lithium tantalate piezoelectrical lab-on-a-chip sensor with a concentric circular IDT as an actuator along with TB-IDT and focused IDTs (FIDTs) for effective sensing is demonstrated. The piezoelectrical wafer 36°YX-lithium tantalate is known to generate shear horizontal (SH) waves along the wafer’s X-direction and Rayleigh waves along the orthogonal directions while saturating a significant part of the thickness of the wafer [39]. The sensor platform is incorporated with a waveguide layer, which results in the formation of Love waves, which are very sensitive to the slightest mass loading [18,40,41,42,43]. The transversely polarized waves, or SH waves, have limited normal displacement on the surface of the wafer and thus do not decay under liquid media. This is highly desirable, especially in the fields of biosensing and chemical sensing [18,40,41,42,43]. The novel TB-IDT has a broader frequency enabling wider frequency range, spectrum access, along with varied amplitude configurations of the electrodes. This study involves the incorporation of the tone burst signal for effective actuation instead of a conventional continuous signal, leading to different actuation frequencies. Additionally, the article presents the computational simulation results that are validated with the experimental sensing results.

## 2. Model-Assisted Design of SAW Sensor

### 2.1. Bulk Wave in 36° YX Cut-Lithium Tantalate

In anisotropic material, the wave characteristics are direction-dependent. Composite materials for aerospace structures, piezoelectrical materials, magnetic materials, biological tissues, and single crystals such as quartz, silicon, etc. are anisotropic materials [44,45,46,47,48]. The wafer involved in this lab-on-a-chip device is piezoelectric, thus it is by default anisotropic in nature. The wavefronts in an isotropic medium elicit into a spherical shape due to the uniformity, and the acoustic velocities are equal in every direction [39]. On the other hand, such is not the case for our piezoelectric wafer. The acoustic wave velocities are non-uniform, which results in different intensities along different directions [39].

Solving Christoffel’s equation in an anisotropic medium will result in the direction-dependent modal wave velocity of different wave modes like quasi-longitudinal (qL), quasi-fast shear (qFS), and quasi-slow shear (qSS) wave modes. The governing differential equation, or Navier’s equation [49], in any medium without a metalized electrode (without piezoelectric coupling), could be written as
(1)$$\sigma_{ij,j}-\rho (x_{j})\ddot{u_{i}}=0\;or\;\nabla .\sigma =\rho (x)\ddot{u}$$
where $\sigma_{ij}$ is the stress and $u_{i}$ is the displacement at a point ($x=x_{i}{\hat{e}}_{i}$) in a bulk anisotropic solid. Substituting stress $\sigma_{ij}$ with constitutive equation $\sigma_{ij}=C_{ijkl}(x_{p})e_{kl}$, and strain with displacement functuons as $e_{kl}=\frac{1}{2}\left(u_{k,l}+u_{l,k}\right)$. In Equation (1), the elastodynamic equation will read
(2)$$C_{ijml}\frac{\partial^{2}u_{m}}{\partial x_{j}\partial x_{l}}+F_{i}=\rho {\ddot{u}}_{i}$$

To monochromatically solve the wave propagation problem, let us assume monochromatic displacement potential function as
(3)$$u_{m}=Ap_{m}e^{i(k.x-\omega t)}$$
where *A* is the scalar wave amplitude. Monochromatic wave frequency is denoted as ω. The wave vector is **k**, and **x** is the position vector. k.x is the dot product between **k** and **x** represents the phase accumulation. $\hat{p}=p_{m}{\hat{e}}_{m}$ is called the polarization vector. It is a unit vector pointing to the direction of the particle motion of the respective wave modes. After substituting Equation (3) in Equation (2) and following a few mathematical simplifications the Navier’s equation can be written as
(4)$$\left[C_{ijml}k_{j}k_{l}-\rho \delta_{im}\omega^{2}\right]Ap_{m}=-F_{i}$$

Without the body force, the homogeneous Navier’s equation in anisotropic media will result in Christoffel’s equation. Using Christoffel symbol, the equation can be written as
(5)$$\left[C_{ijml}n_{j}n_{l}-\rho \delta_{im}c^{2}\right]p_{m}=0\;or\left[\Gamma_{im}-\rho \delta_{im}c^{2}\right]p_{m}$$

This is an eigenvalue problem, where $c^{2}=\omega^{2}/k^{2}$ is the square of the phase velocity of a wave mode along the direction of k vector. The $n_{j}$ parameters are the direction cosines of the unit vector along the wave propagation direction (along the k vector). The solution of this equation will provide three eigenmodes with wave velocities, namely quasi-longitudinal or $c_{qL}$, quasi-fast shear or $c_{qFS}$ and quasi-slow shear $c_{qSS}$, respectively. Figure 1 shows the 3D view of the velocity profile of quasi-shear wave mode in 36° YX cut-lithium tantalate.

Using lithium tantalate, the anisotropic material properties (with a density of 7450 kg/m^3^) in Equation (6) and the eigen solutions at 10 MHz were found for all possible wave vector directions. These are shown in Figure 1 and Figure 2. The material properties were taken from the COMSOL Multiphysics Material Library; however, they could also be found in [50]. Material properties include the elastic constitutive matrix as **C** in Stress-Charge form, the piezoelectric coupling matrix, and the relative permittivity matrix in Stain-Charge form as **D** and **E**, respectively.
$$C=\left[\begin{array}{cccccc} 232.966 & 46.8904 & 80.2342 & -11.0267 & 0 & 0\\ & 232.966 & 80.2342 & 11.0267 & 0 & 0\\ & & 275.364 & 0 & 0 & 0\\ & & & 93.8995 & 0 & 0\\ & Sym & & & 93.8995 & -11.0267\\ & & & & -11.0267 & 93.038 \end{array}\right]GPa;$$
(6)$$D=\left[\begin{array}{cccccc} 0 & 0 & 0 & 0 & 26 & -14\\ -7 & 7 & 0 & 26 & 0 & 0\\ -2 & -2 & 8 & 0 & 0 & 0 \end{array}\right]\times 10^{-12}\;C/N;$$
$$E=\left[\begin{array}{ccc} 51 & 0 & 0\\ 0 & 51 & 0\\ 0 & 0 & 45 \end{array}\right]$$

The piezoelectric wafer in this device has a defined thickness along the Y-axis. This thickness generates guided-wave modes, which are to be used on the XZ-plane. Thus, the velocity profile of the piezoelectric wafer is constructed in the XZ plane. Figure 2 shows the velocity profile of the three wave modes in the 36°YX cut-lithium tantalate.

The quasi-fast shear spawned the formation of the velocity profiles dominating at 0° and 45°, corresponding to the X-axis. A low velocity can be observed along the 90° direction which is along the Z-direction, also the direction of polarization. Although bulk wave modes are suitable for understanding, it is necessary to understand the Guided wave modes in 36° YX cut-lithium tantalate before the design is conducted.

### 2.2. Guided Wave in 36° YX Cut-Lithium Tantalate

A numerical finite element model of the 36° YX cut-lithium tantalate wafer with 350 µm thickness is created in COMSOL Multiphysics software version 4.6 to understand the guided wave modes near 10 MHz frequency. As the frequency is high and the finite element model requires a minimum of 10 elements per wavelength to have satisfactory convergence, a small 5 mm by 5 mm wafer is considered. To simulate the infinite wafer, assuming periodic boundary conditions in both directions, Bloch-wave theory is exploited in the model [49]. A rotated coordinate system was selected for the lithium tantalate wafer, having Eulerian angles corresponding to α = 0°, β = −54°, and γ = 0° for the simulation and transforming it to a 36° YX cut profile. Frequency eigen solutions were found between zero and π/a, where a is the periodicity of the wafer (5 mm). As per Bloch wave theory, it is known that any wave number solution k along any direction has an additional similar solution at $k+\frac{2\pi n}{a}$. After finding the numerical solution along $k_{x}$ and $k_{y}$, only real eigenvalues near 10 MHz were collected, followed by a few modes that have very low (10^−9^) imaginary parts. Figure 3 and Figure 4 show a few of those real mode shapes that provide evidence that the wafer may dominate shear wave modes near 10 MHz along the x-axis and at 45°, 135°, and 180° with respect to the x-axis. It also shows that the Rayleigh wave may dominate along the Z-axis. For the direction convention of the SAW wafer, please refer to Mandal et al. 2022 [18]. For simulation, the following material parameters were used. The constitutive matrices are written in Equation (6).

Convergence was assured through discretizing the wafer with the minimum size of the elements to be 10 times smaller than the smallest possible wavelength in the wafer up to 12 MHz, which is greater than 10 MHz. From all the modes in Figure 3 and Figure 4, it is evident that the number of waves that fit within the Bloch structure of 5 mm dimension is always an integer multiple of 3, 4, 6, 12, or 24. Next, to commensurate the reported shear wave velocity in the literature and commercial vendor-provided wave velocity in 36° YX cut-lithium tantalate [51,52,53], with the model-based findings, 12 numbers were selected for best-optimized sensing. Hence, a wave length of 416 µm was decided, which is 12 divisions of 5 mm. If a 416 µm wavelength is used to construct the interdigitated electrodes (IDTs), all the above modes in Figure 2 should be sensed by the wave in different directions.

Based on the modal analysis of the wafer for guided wave modes, the design of the sensor test bed was conducted. It is proposed that the sensing sites be placed at 0°, 45°, 90°, 135°, and 180° with respect to the X-axis. This enables the placement of sensing test beds along these respective directions, as different directions have different impacts on acoustic wave velocity and acoustic modes due to the anisotropic nature of the substrate. As a result, focused IDTs (F-IDTs) were introduced to 0°, 45°, 135°, and 180°. As a Rayleigh wave was observed along the 90° (Z-direction), the tone burst IDT (TB-IDT), being the multifrequency and multi-amplitude accessing electrode, was placed in this direction. Except for TB-IDT, which required special treatment discussed in Section 3.2, all F-IDT used the design parameter of wavelength equal to 416 µm found above. Sensor fabrication materials and methods are discussed in the following sections.

### 2.3. Design of the Sensing Platform and Computational Simulation of the Sensing Platform

The SH wave-based sensing platform is proposed as an alternative approach for higher and better sensitivity using multidirectional and multifrequency accessibility. Figure 5 shows a concentric circular IDT, which is designed as an actuator at the center of the platform. The main purpose of the concentric circular IDT is to produce omni-directional wavefronts within the same plane to access different directions of wave propagation, which is evident from Section 2.2. Omni-directional wavefronts due to the anisotropy of the wafer provide other possibilities to generate and extract different wave features in different directions. Thus escalating the chances of higher sensitivity.

The concentric circular IDT was designed based on a 10 MHz central frequency. Using the guiding Equation (2), the wavelength was derived.
(7)$$f=\frac{c}{\lambda }$$

Here, f is the central frequency, c is the acoustic wave velocity, and λ is the wavelength of the wave. The acoustic wave velocity of the 36° YX cut-lithium tantalate wafer is considered 4160 m/s, as reported by others [51,52,53] and to be consistent with the modeling results presented in Section 2.2. Using the equation above, the wavelength is derived at 416 μm, which is the same as the wavelength obtained from modeling and analysis. As per the design, the concentric circular IDT consists of a positive and a negative terminal with two similar width spacings within one pitch of the IDT. Hence, the width and spacing of the concentric circular IDT were 1/4th of the wavelength, which is 104 μm. Figure 5 shows the schematic and design details of the concentric circular IDT as the actuator.

The concentric circular IDT is adjacently surrounded by the five sensing test beds, which are located at a 15-wavelength distance from the periphery. In a similar fashion, the sensory interdigitated electrodes are placed at 30 wavelengths at an angular configuration of 0°, 45°, 90°, 135°, and 180°, respectively. The whole schematic of the sensing platform is mirrored on the other opposite half, and thus making 10 testing sites for better reproducibility and selectivity option exploration. Figure 6 shows the schematic and the actual image of the sensor on top of the piezoelectric wafer.

### 2.4. Wave-Field Modeling in 36° YX Cut-Lithium Tantalate

For wave-field computation, COMSOL Multiphysics was employed to visualize the wave field that could be generated and sensed in the designed wafer. A concentric circular IDT was selected as the actuator. It was previously hypothesized that the concentric circular IDT is capable of producing omni-directional wavefronts in the same plane. To verify, a concentric circular IDT with a few electrode fingers was modeled on top of a 36° YX cut-lithium tantalate wafer (refer to Section 2.2 for material properties) having a thickness of 350 µm. Like before, a rotated coordinate system was selected for the lithium tantalate wafer, having Eulerian angles corresponding to α = 0°, β = −54°, and γ = 0. The concentric circular IDT was applied with a 10 V AC input. The displacement field was computed in the time domain at each time step. Figure 7 displays the concentric circular IDT producing acoustic wavefronts in a 36° YX cut-lithium tantalate wafer.

The piezoelectric wafer is anisotropic in nature, and thus, it is expected that the wave velocities would be different along different directions, resulting in the formation of distorted circular wavefronts. The same simulation results also validated the formation of shear horizontal waves. As the shear horizontal waves are non-decaying in nature along the thickness direction, the wave seems to saturate the full thickness of the wafer. Figure 8 displays the simulated results of the backside of the wafer. The waves can be seen transmitting along the thickness and all the way to the bottom of the wafer.

After the confirmation of the shear horizontal waves and validity of the concentric circular IDT, the different sectors of the biosensing platform were simulated for acoustic sensing. Due to the computational system’s limitation, each configuration was simulated separately. This article demonstrates the surface acoustic waves-based actuation and sensing of the tone burst configuration, situated at 90°. Figure 9 shows the wave propagation simulation (actuation and sensing). The creation of tone burst interdigitated electrode pattern is discussed in Section 3.

The simulation was performed by actuating the concentric circular IDT using 10 Vpp AC input with a tone burst signal with a 10 MHz central actuation frequency. The simulation model platform was surrounded by a low-reflecting boundary layer that consists of the width of twice the wavelength of the acoustic wave. This boundary layer was added to prevent boundary reflections of the waves (from the geometry boundary) from being as high as possible. The time domain simulation was performed for 8 microseconds, with each time step of 4 nanoseconds for better simulation convergence. The surface displacement field throughout the simulation demonstrated the propagation of the acoustic waves. The middle of the sensing model consisted of a circular sensing membrane with a thickness of 1 micron and a 1000 micron radius. A sensory voltage probe was attached to the tone burst IDT for sensing. Due to the converse piezoelectric phenomenon, the acoustic waves as they propagated to the tone burst IDT resulted in the generation of electric potential, which is then sensed. In a similar fashion, the other configurations of the sensing electrodes, which are sensory FIDTs at 0°, 45°, 135°, and 180°, were simulated, and the electrical output voltage waveform was recorded. In addition, a FIDT placed in 90° configuration was also tested for acoustic wave phenomena and signal features. Figure 10 displays the signal waveform recorded by the sensory IDTs at different configurations.

The simulation results clearly indicate the formation of new wave packets in addition to the cross-talk signal, which is the trigger point. The formation of the new wave packets is due to the phenomena of wave reflection and dispersion. The packets consist of meaningful and sensitive information which are formed primarily due to the different wave modes and the concept of interferences from different geometries and directions. These waveforms are analyzed using numerous signal transformation and acoustic feature extraction tools for quantification. The simulation results confirmed the formation of new wave packets and wave dispersion phenomena. In the next step, the sensors were fabricated and verified for the signal dispersion and formation of wave packets.

## 3. Materials and Methods

### 3.1. List of Instruments and Materials

The 36° YX cut-lithium tantalate wafers were purchased from *Custom Glass and Optics, USA*. S1813 positive photoresist was purchased from *Kayaku Advanced Materials, Westborough, MA, USA.* The photomask was fabricated at *FineLine Imaging, Colorado Springs, USA.* The etching solutions were purchased from *Transene Co, Danvers, MA, USA.* All the other fabrication tools and chemicals were used at the *Chapel Hill Analytical and Nanofabrication Laboratory (CHANL), University of North Carolina at Chapel Hill, USA.* The computational and electrical measurements and analysis were performed at the *Integrated Material Assessment and Predictive Simulations laboratory (iMAPS), University of South Carolina, USA*.

### 3.2. Tone Burst Interdigitated Electrodes Design

A tone burst signal is a simple sinusoidal signal windowed with a Gaussian function. The guiding equation of the tone burst signal is given by
(8)$$X\left(t\right)=\sin\left(N\;*\;2\pi ft\right)\;*\;e^{{\left[-\frac{\left(t-T0\right)}{\frac{T0}{2}}\right]}^{2}}$$
where *N* is the number of cycles, *f* is the central frequency, *t* is the time and *T_0_* refers to the time-period of the burst signal [33]. The tone burst signal is a modulated wave packet. Unlike a continuous signal, the tone burst is widely used in structural health monitoring and non-destructive evaluation of materials and structures [39,54]. Generally, in surface acoustics, the propagated waves interact with different geometries, specimen types, and boundaries, and the dispersion results in the formation of additional wave packets. The newly formed wave packets are very sensitive and crucial for quantification. This phenomenon leads to the formation of more sensitive coda waves compared to conventional continuous signals and is thus hypothesized to improve sensitivity. Figure 11 shows a 5-count tone burst signal operating at 2.5 MHz central frequency and its frequency transformation.

To construct the tone burst electrode, the design was assembled at a 2.5 MHz central frequency. The maximum amplitude can be observed at 2.5 MHz, which is the central frequency. Other frequencies can be observed adjacent to the central frequency. All the frequencies on the right corresponding to the central frequency are the higher frequencies, and the left side represents the lower frequency content. For the actual sensor design, frequencies ranging from 1.5 MHz to 3.5 MHz were selected. All the frequencies have different amplitudes. A total of 15 pairs of electrodes were designed within the frequency range selected. The width and height of the electrodes are derived based on the frequencies and their corresponding amplitudes. The one-fourth wavelength corresponding to each frequency was derived as the individual width of the interdigitated electrode. For the aperture, or the height of the individual electrodes, the frequency domain curve was normalized with respect to the maximum peak amplitude. Under the curve, a 20-times wavelength span corresponding to all the respective frequencies was demonstrated for the half aperture. Later, the aperture length was doubled to construct the final tone burst IDTs. Figure 12 shows the schematic design of the 5-count tone burst IDTs.

### 3.3. Sensor Fabrication

The sensing platform was fabricated in multiple steps. In the first step, a 1.2-micron-thick SiO_2_ layer was deposited via plasma-enhanced chemical vapor deposition (PECVD) onto the bare 100-mm, 36° YX cut-LiTaO_3_ SAW-Grade wafer (*Custom Glass and Optics, Williamsburg, VA, USA*). The PECVD was performed at 200 °C, 25 W plasma power, and 800 mTorr using 2% Silane (balanced He) and N_2_O. Additional care was taken to avoid excessive stress and possible cracking of the piezoelectric wafers by slowly ramping up and down the chamber temperature. A temperature change of less than 10 °C/min was sufficient to avoid damaging the wafer. This ramping rate was also used for subsequent photoresist baking steps.

The SiO_2_ was patterned using photolithography and wet etching and comprises the first layer of the Au/Cr/SiO_2_ test beds. Patterning this layer was performed by spin coating a photoresist adhesion promotor (20% hexamethyldisiloxane (HMDS) and 80% propylene glycol monomethyl acetate (PGMA) solution) at 3000 rpm for 40 s. S1813 positive photoresist (*Kayaku Advanced Materials, Westborough, MA, USA*) was then spin coated at 3000 rpm for 40 s and baked at 110 °C for 1 min. The photoresist was exposed to UV light with a dose of 150 mJ/cm^2^ using a laser-printed photomask (*FineLine Imaging, Colorado Springs, CO, USA*). The exposed resist was then developed in MF-319 developer (*Kayaku Advanced Materials*) for 2 min, rinsed with deionized water, and dried with N2.

Once the lithography process was completed, the resist pattern was transferred into the underlying 1.2-µm SiO_2_ layer by etching in a buffered oxide etchant (BOE) 10:1 solution. The etch rate of the oxide layer was approximately 300 nm/min. After defining the first layer of the SiO_2_ test beds, a second layer of SiO_2_ was deposited on the same wafer. In this layer, a 50-nm-thick SiO_2_ layer was patterned using the lithography/etch procedure previously described. This SiO_2_ layer acts as a mask for the aperture of the tone burst interdigitated transducers (TB-IDTs) that are defined in the next fabrication step.

The TB-IDT and FIDTs and the second layer of the test beds (sensing membranes) were fabricated via e-beam evaporation, followed by photolithography, and finally wet etching. E-beam evaporation consisted of depositing a 10 nm Cr adhesion layer followed by 100 nm of Au. The lithography process was identical to the process described previously. It should be noted that alignment of the IDE photomask to the 50 nm SiO_2_ aperture (for the TB-IDT) is critical and hence was taken care of before exposure to the photoresist. Au/Cr patterns were finally defined via wet etching with an I2/KI Au etch solution (*Transene Co., Danvers, MA, USA*), followed by a nitric acid/ceric ammonium nitrate Cr etch solution (*Transene Co.*). The etch rates of the Au and Cr solutions were approximately 28 A°/s and 40 A°/s, respectively. Figure 13 shows the schematic of the fabrication process.

### 3.4. Experimental Process

The sensing platform is based on simultaneous actuation and sensing mechanisms. For actuation, the concentric circular IDT, which is built at 10 MHz central actuation frequency (4 times the TB-IDT central frequency) and assuming wave velocity in the substrate of 4160 m/sec, was excited using a 5-count tone burst signal from a digital arbitrary function generator (Tektronix AFG 31000, Beaverton, OR, USA). An input of 10 Vpp signal amplitude was applied throughout the experiments. For the sensing purpose, we incorporated a Digital Oscilloscope (Tektronix MDO3024 with 200 MHz, 2.5 GS/s) with a data acquisition of peak averaging 512 samples. The sensor terminals were connected using a four-point probe station. Figure 14 demonstrates the actual experimental setup.

In the first stage, different configurations of the interdigitated electrodes, i.e., tone burst IDT and F-IDTs, were tested for the tuning of the resonant frequency. The piezoelectrical wafer is essentially an anisotropic material, and hence, different directions are expected to have different phase velocities with different modal dominance. Additionally, different geometrical configurations of the interdigitated electrodes and boundaries result in different resonating frequencies and the formation of different acoustic wave modes. Thus, all the different configurations of the electrodes were sensed for resonance by exciting the actuator electrode into a vast range of frequencies based on their tuning responses. The process is referred to as ‘*frequency tuning*’ corresponding to each configuration of the interdigitated electrodes. Figure 15 shows the direction-dependent frequency tuning curves. The sensing platform consists of a concentric circular IDT as the actuator and the TB-IDT and F-IDTs as the sensors. The concentric circular IDT was excited at the resonating frequency (after tuning) with the input tone burst signals from the function generator, and the corresponding sensory signals were captured from F-IDTs and TB-IDTs using the digital oscilloscope. The signals were frequency transformed and analyzed, as discussed in the next section.

## 4. Results and Discussion

### 4.1. Actual Signal Analysis of the Sensing Platform

The sensing platform was tested for all the individual IDTs (4 F-IDTs and 1 TB-IDT). As each electrode is placed in different directions of the piezoelectric wafer, it is evident that the acoustic velocity will be different in these directions from the same point of actuation. As a result, the frequency gets affected due to different directions, wave modes, and geometry, due to which the resonating frequency showing maximum dispersion corresponding to each IDT is different. Each configuration of the IDT was tuned to its resonating frequency, and data was recorded. Figure 16 shows the actual experimentally captured signals of the sensors at different configurations in their tuned resonating frequencies.

As mentioned previously, all the IDTs were swept with actuation frequencies ranging from 100 kHz to 40 MHz. It is evident that all the IDTs, irrespective of their position and configuration, resulted in the formation of new wave packets. As hypothesized from the simulation, the hypothesis was validated as the formation of the new wave packets can be observed in all the IDTs’ sensory signals. As depicted in the simulation results, a similar trend was observed where the waveforms of the 0° and 180° configurations are similar. Such a phenomenon was also observed in the case of the computational simulation. The 0° and 180° configurations resulted in optimum frequency tuning at 10 MHz central frequency with the highest first wave packet amplitude after the cross-talk. Additionally, the F-IDTs at 45° and 135° were tuned to 12.5 MHz, showing their highest amplitude after the cross-talk.

The tone burst IDT was tuned at two different frequencies: 15 MHz and 35 MHz. Different wave modes were observed at these respective actuation frequencies. The wave packet profiles are very discrete at the mentioned frequencies. To understand this phenomenon, the frequency transformation of the time domain signals was modeled. Figure 17 displays the frequency transformation of the above time domain signals.

It can be clearly seen that all the electrodes (F-IDTs and TB-IDT) resulted in a meaningful frequency plot. All the plots displayed a sharp peak amplitude near their respective actuation frequencies. The concentric circular IDT, as mentioned earlier, was designed for this particular frequency and can be seen dominating in the frequency spectrum. An interesting trend was observed from the frequency plot. Although the excitation frequency was 10 MHz, different electrodes resulted in different tuning frequencies. In the case of both 0° and 180° configurations, a significant peak can be observed at their tuned central actuation frequency of 10 MHz. In both cases, the frequency waveform pattern is similar. However, 180° configurations resulted in lower amplitude. In addition, both configurations showed the formation of an additional significant peak at near 10.82 MHz. An interesting trend was observed in the cases of the 45° and 135° configurations. Although these electrode configurations were tuned to their maximum response at the 12.5 MHz central actuation frequency, their peak amplitude in the frequency spectrum is observed at nearly 14.3 MHz. A natural frequency shift is present in these configurations. Additionally, a lower-amplitude adjacent peak near the central peak corresponding to these two configurations can be seen at 14.8 MHz. The tone burst IDT was tuned at 15 MHz and 35 MHz. Corresponding to both frequencies, the tone burst IDT signal resulted in the formation of multiple peaks along the frequency domain. At 15 MHz actuation, significant frequency peaks are displayed at 16 MHz. When zoomed into the range of actuation frequency of 15 MHz, the formation of many peaks can be observed, as shown in Figure 17.

Moreover, many other significant peaks were observed in different frequency ranges. At an actuation frequency of 35 MHz, a significant number of peaks can be observed in addition to the central actuation frequency. Significant frequency peaks are formed at higher frequencies at 93.5 MHz, 96.7 MHz, and 130 MHz. Additionally, smaller amplitude peaks can be observed at 50 MHz, 160 MHz, and 235 MHz. It is evident from the frequency domain that multiple frequencies, in addition to the central frequency, are accessible using the tone burst excitation signal and sensing. In the case of tuning response at 35 MHz central actuation, similar formations of multiple frequency peaks were observable. Here as well, significant frequency peaks are formed at higher frequencies, which are at 93.5 MHz, 96.7 MHz, 130 MHz, and 215 MHz. The lower amplitude peaks can be observed at 50 MHz, 76 MHz, 123 MHz, and 172 MHz. Additionally, multiple peaks can be observed at 38 MHz, which is closer to the central actuation frequency as shown in Figure 17. From the experimental data, it is evident that the unique tone burst IDT was capable of generating multiple frequencies in addition to its main central actuation frequency. Although expensive to fabricate, tone burst IDT provides more flexibility compared to focused IDT in selecting multiple frequency peaks simultaneously for better detectability. This potential should be explored for future applications in frequency shift-based detection methodologies.

### 4.2. Example Actual Sensing Using Microcystin-LR

In this section, an actual example of the functionality of the sensor is validated. Out of many future potential sensing possibilities, such as gas sensing, chemical sensing, and RF sensing; the biosensing was selected as an example for the validation of the sensor in this study. One of the leading water contaminants, Harmful Algae Blooms (HAB) of cyanobacteria occur frequently in bodies of water that are lethally harmful [55]. Cyanobacterial algae blooms often generate undesirable color, odor, and taste, but very importantly, lethal toxins such as microcystins, neurotoxins, hepatoxins, and dermatoxins. In this study, one of the lethal toxins (Microcystin-LR) is detected using the proposed sensing device as an example.

#### 4.2.1. Functionalization of the Sensing Test Sites

The sensor platform was washed multiple times with Acetone, IPA, and DI water. Then, the sensing test site was incubated with 100 mM Thiourea overnight in order to develop a Self-Assembled Monolayer (SAM) over the gold surface. After the incubation, the sensor surface was cleaned using Ethanol and DI water to remove the excess SAM. In the next step, the surface activation was made at the sensor test site using 75 mM N-(3-Dimethylaminopropyl)-N′-ethylcarbodiimide hydrochloride (EDC), followed by 50 mM N-Hydroxysuccinimide (NHS) complex. Following this step, 50 μg/mL Microcystin-LR (MC-LR) antibodies in Phosphate Buffer Saline (PBS) pH 7.4 were incubated overnight at 4 °C on the sensor surface. After the incubation process, excess non-binded antibodies were removed by washing away with PBS, and then the non-reactive surfaces were blocked using Ethanolamine. In the final step, the MC-LR antigens at a 1 μM concentration were dispensed on top of the functionalized sensor surface.

#### 4.2.2. Comparison of Sensor Signals: MC-LR Antibodies and Antigens for the Detection

The experimental setup used for this study is the same as referred to in Section 3.4. MC-LR antigens were added to the functionalized sensor surface, and data was recorded for 55 min (to see the stability of detection) with an interval of 5 min at each step. The recorded signals were transformed for detection purposes. The envelope obtained from the Hilbert transformation of the signals in the frequency domain was plotted.

Figure 18 shows the comparison of the signals in the frequency domain before and after the addition of MC-LR antigens on top of the antibody-immobilized sensor.

Results shown in Figure 18 indicate that, in addition to the central actuation frequency (10 MHz), different other frequencies (both higher and lower) show significant peak shifts. At a 30 min time interval, a maximum central frequency shift of 125 kHz can be observed. Figure 19a shows the zoomed peak shift difference recorded in Figure 18. Similarly, at the higher frequency lobe (near 11 MHz), another peak shift of 125 kHz can be seen at a 20 min time interval after the addition of antigens (Figure 19b). Additionally, at a 50 min time interval, the highest frequency shift of 325 kHz is recorded at near 9 MHz central actuation. In a similar fashion, significant other peak shifts can be observed (in the range of kHz) throughout the frequency domain over time, thus proving the hypothesis of the broader bandwidth and better sensing capabilities of the sensors. An interesting trend was observed in which the amplitude of the signal with antigens decreased as time elapsed. The proposed sensor clearly shows an enhanced range of frequency shifts (in kHz) in addition to a broader spectrum of detection and does not just rely on a single central actuation frequency shift or a single detection parameter. Thus, it enables a better sensing platform for different applications and gives it a leading edge over conventional SAW-based sensing and its methodologies.

## 5. Conclusions

In this article, we demonstrate tone burst-based sensing and actuation on a lab-on-a-chip device that has access to multiple frequencies and multiple directions simultaneously. The article is focused on providing the materials and methods to create the TBIDT for more generalized application to multiple fields of sensing. Hence, a specific demonstration of biosensing was not presented. It is proven that by employing TBIDT, the sensing platform was able to generate multiple frequency peaks. They are useful for several applications beyond biosensing, for example, chemical sensing, gas sensing, microfluidics, and RF sensing, etc.

By incorporating a single concentric circular IDT, the device was built to activate multidirectional wavefronts. The non-dispersive, non-decaying shear horizontal wave modes are dominant in the device, which makes it highly sensitive to mass loading and suitable for their use in liquid media. The modeling of the bulk wave modes and the guided wave modes along the plane of a piezoelectric wafer elicited the profiles of the acoustic wave velocity and their respective mode shapes. Different acoustic wave modes and their unique mode shapes played a crucial role in designing the electrodes (sensory IDTs) and the placement of the testing sites. The frequency transformation of a tone burst signal with a 2.5 MHz central frequency helped design the shape of the electrode profile. Next, unique-tone burst IDTs were fabricated. In comparison to the other sensory IDTs (FIDTs), the tone burst IDTs exclusively accessed multiple frequency peaks in addition to the main central frequency. These additional frequency peaks could be explored by different fields for improved sensing and detection.

The piezoelectric wafer is essentially anisotropic, and different directions of the wafer result in different acoustic velocities. The acoustic waves interact with specific mass loads, leading to a change in acoustic velocity, which contributes to the change in frequency. The concentration of different targets (chemical, biological, or gases) leads to a change in frequency and thus, having access to multiple frequencies can increase the chances of detection. In other words, the probability of detection of the target from multiple frequencies is much higher and more reliable than a single parameter-based detection, which used to be a single central frequency. Moreover, multiple directions are explored, and due to the multiple test sites within the same actuator/sensing platform, the statistical validation of the detection can be tested, which can contribute to the better chances of reproducibility and thus lower the chances of false negatives. The lab-on-a-chip platform is a proof-of-concept device suitable for multiple fields for detection purposes. Users from chemical, electrical, biological, and microfluidics fields can utilize this device for both effective sensing and actuation corresponding to multiple different target analytes or substances.

## Figures and Tables

**Figure 1 sensors-24-00644-f001:**
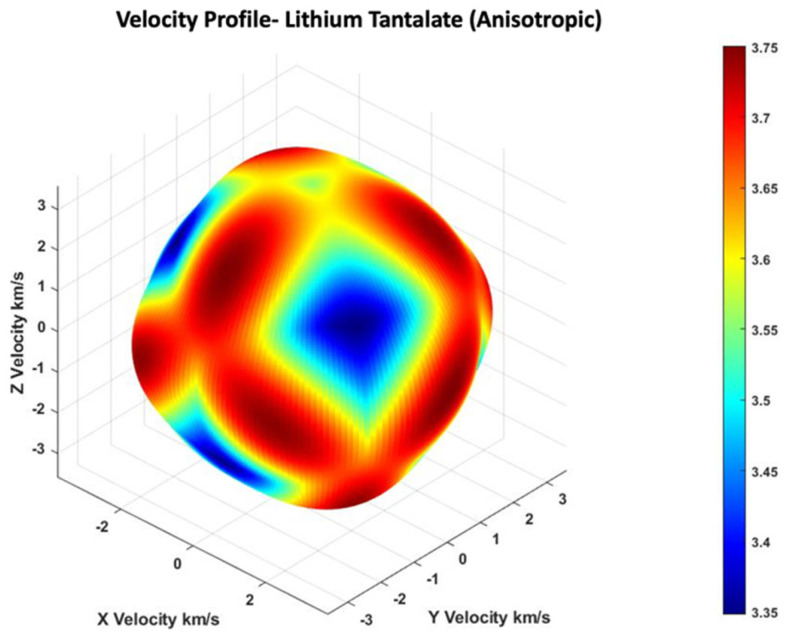
Three-dimensional view of the velocity profile of quasi fast shear wave mode in a 36° YX cut-lithium tantalate.

**Figure 2 sensors-24-00644-f002:**
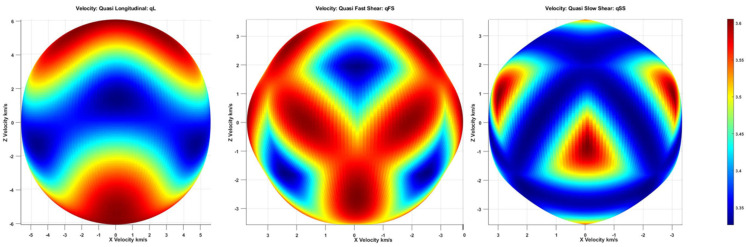
Velocity profiles of the 36°YX lithium tantalate with different modes, qL (**left**), qFS (**middle**), and qSS (**right**).

**Figure 3 sensors-24-00644-f003:**
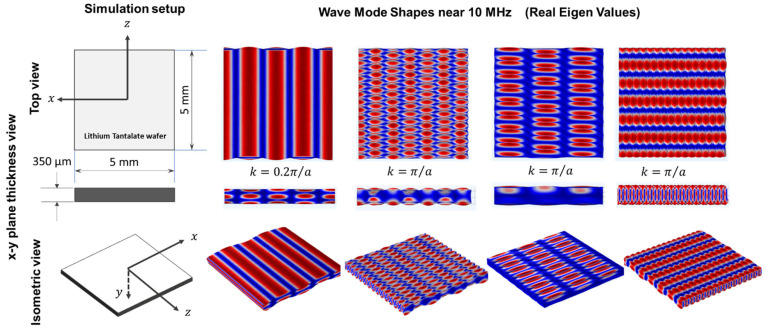
Guided wave mode shapes with real eigenvalues in frequency domain around 10 MHz.

**Figure 4 sensors-24-00644-f004:**
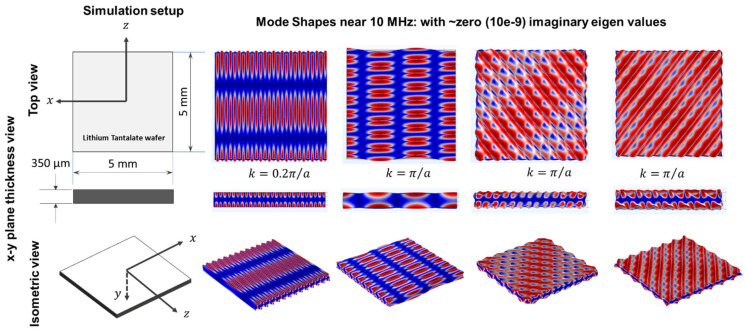
Guided wave mode shapes with eigenvalues in frequency domain around 10 MHz vary with small imaginary parts smaller than 1 × 10^−9^.

**Figure 5 sensors-24-00644-f005:**
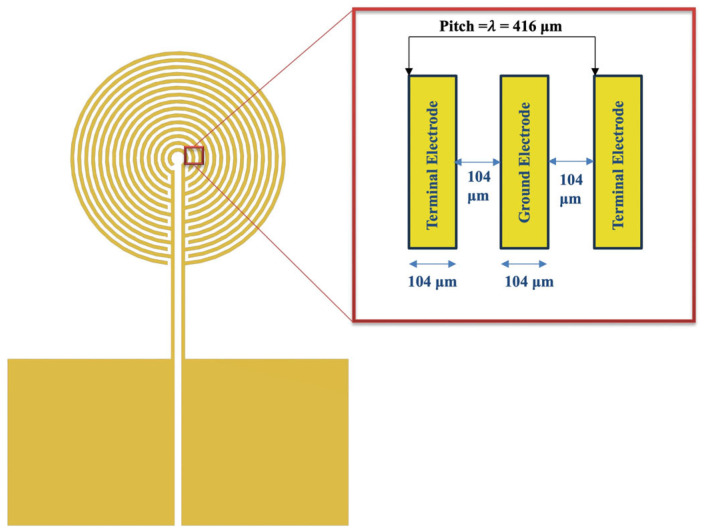
Schematic design of the concentric circular IDT (actuator) for the sensing platform.

**Figure 6 sensors-24-00644-f006:**
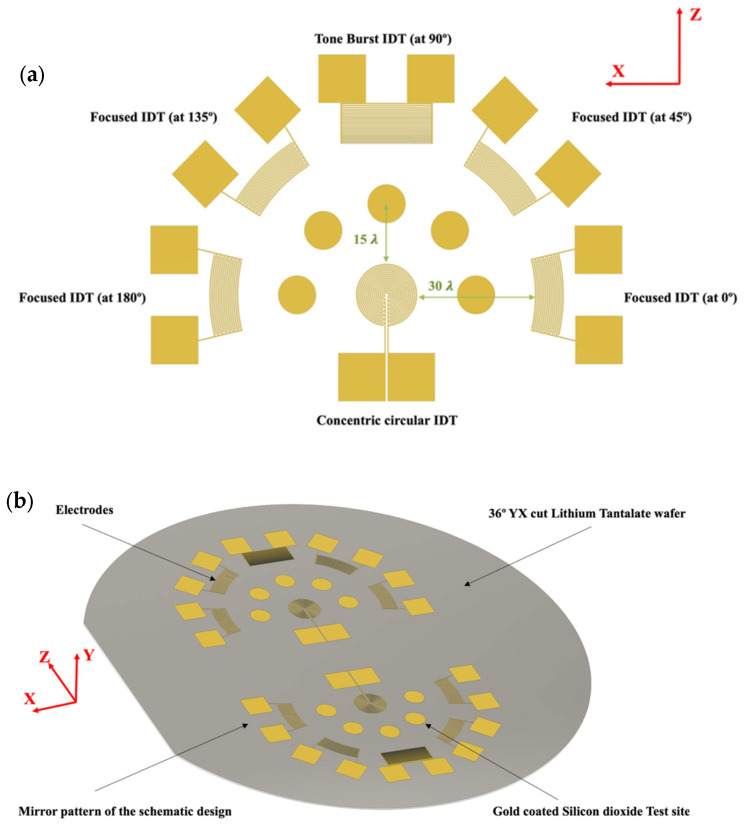
(**a**) 2D-Schematic of the sensing platform, and (**b**) 3D isometric view of the whole sensor.

**Figure 7 sensors-24-00644-f007:**
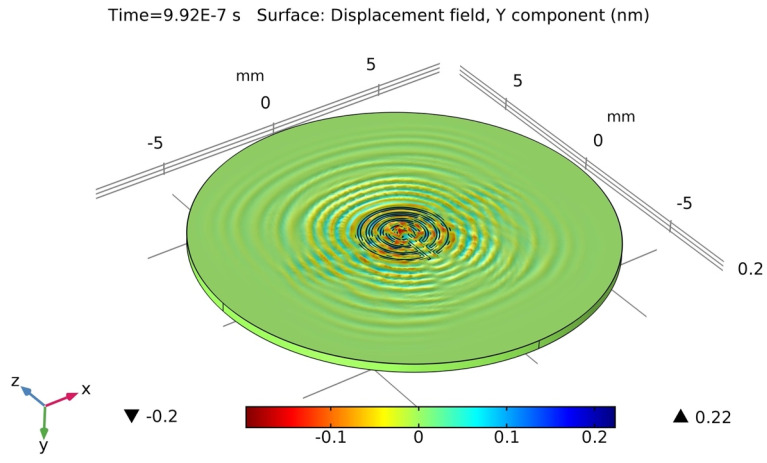
Computation simulation of concentric circular IDT on top of 36°YX lithium tantalate wafer at time t = 0.9927 µs.

**Figure 8 sensors-24-00644-f008:**
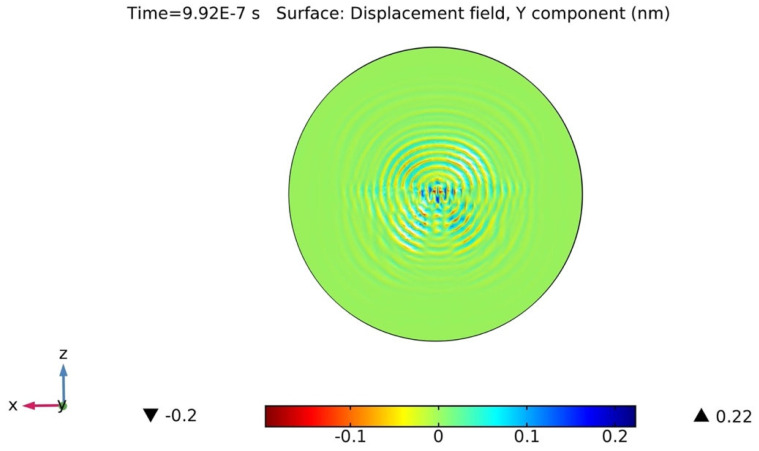
Concentric circular IDT on top of 36°YX lithium tantalate wafer at t = 0.99 µs generating SH waves at the bottom of the wafer (backside). The non-decaying nature of the shear horizontal waves allows waves to transmit along the thickness of the substrate.

**Figure 9 sensors-24-00644-f009:**
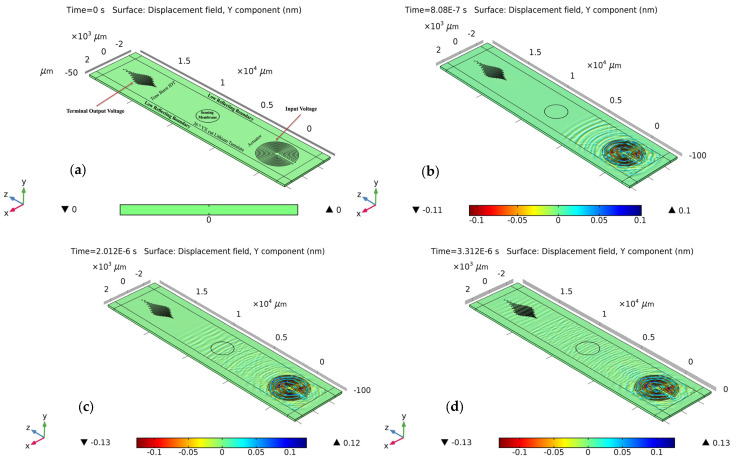
Simulation of the shear horizontal waves generated by the concentric circular IDT and waves propagating along the direction at 90° at time: (**a**) 0 μs, (**b**) 0.8 μs, (**c**) 2.012 μs, and (**d**) 3.312 μs.

**Figure 10 sensors-24-00644-f010:**
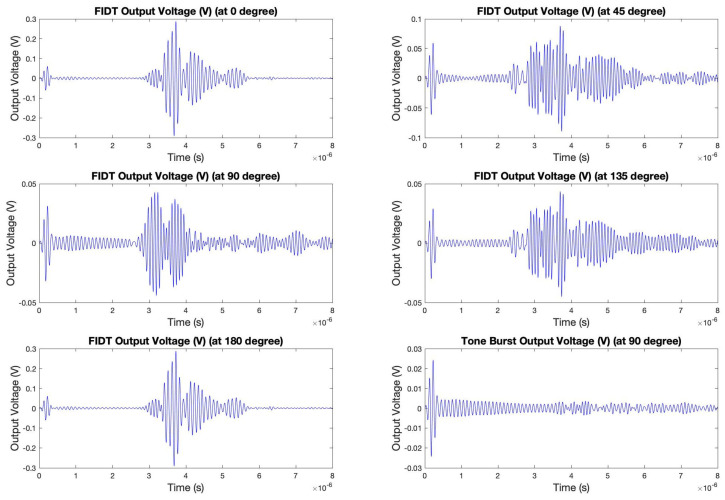
Simulated sensing voltage recorded over time by interdigitated electrodes (Focused and Tone Burst) at 0°,45°, 90°,135°, and 180°, corresponding to the concentric circular actuator using the input tone burst signal.

**Figure 11 sensors-24-00644-f011:**
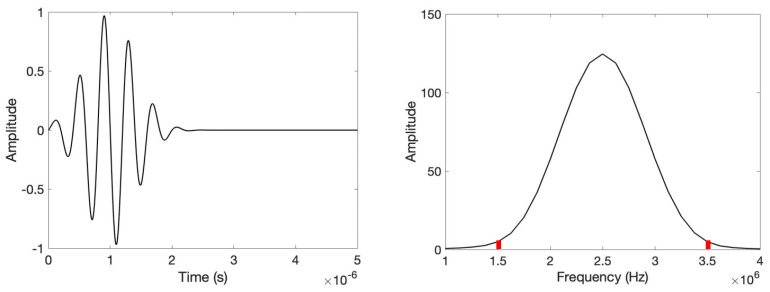
Five-count tone burst signal at 2.5 MHz central frequency (**left**), and frequency transformation of the signal (**Right**).

**Figure 12 sensors-24-00644-f012:**
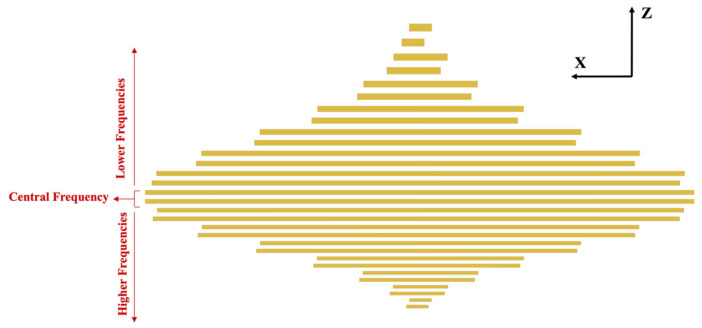
Schematic of the tone burst interdigitated electrodes.

**Figure 13 sensors-24-00644-f013:**
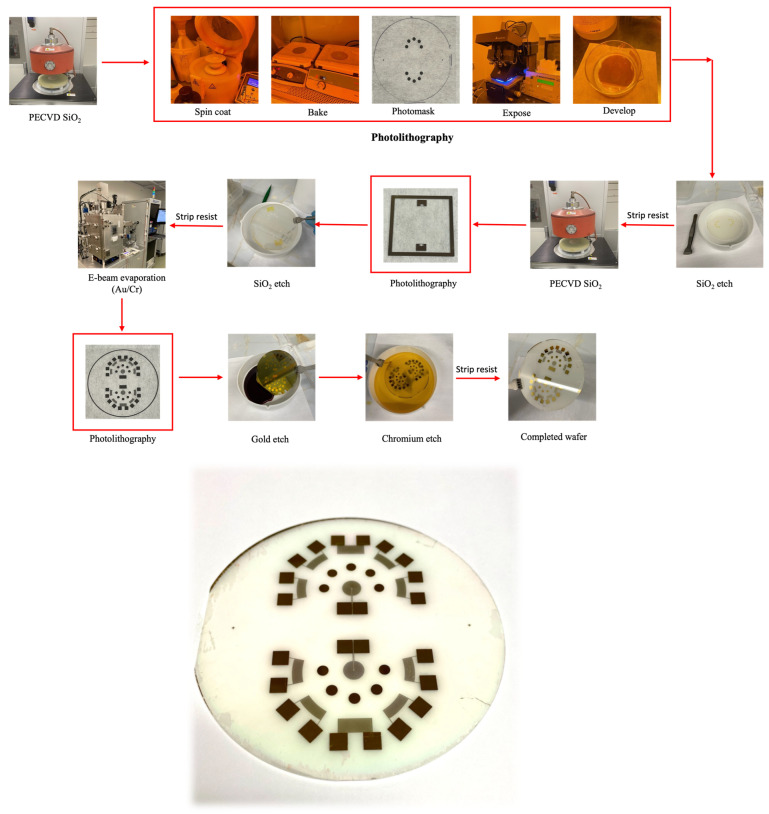
Block diagram of the fabrication process of the sensor (**top**), and actual image of the sensor (**bottom**).

**Figure 14 sensors-24-00644-f014:**
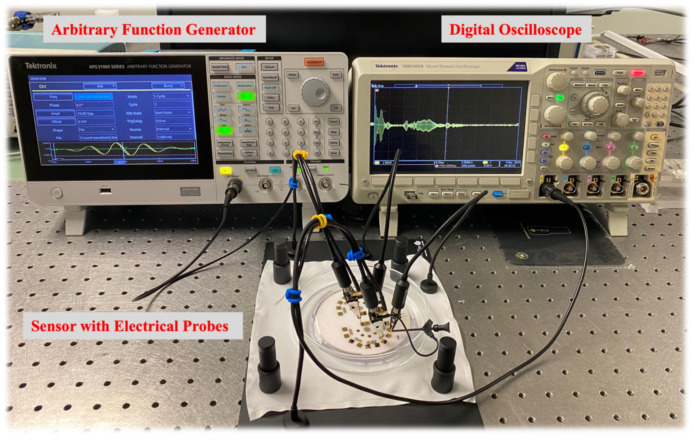
Experimental setup of the sensor and its different components. The image displays the signal response from 135° configuration excited at 12.5 MHz actuation.

**Figure 15 sensors-24-00644-f015:**
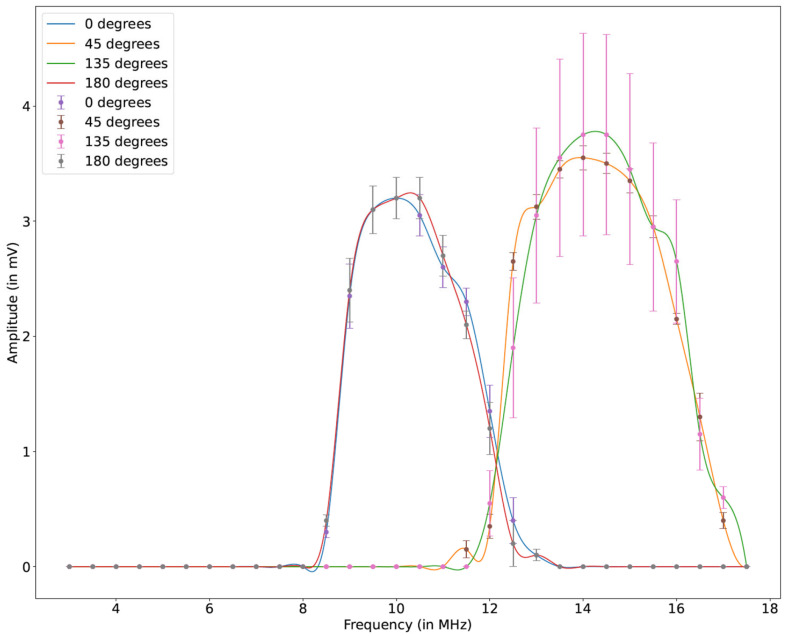
Direction-dependent frequency tuning curves of the F-IDTs at different configurations.

**Figure 16 sensors-24-00644-f016:**
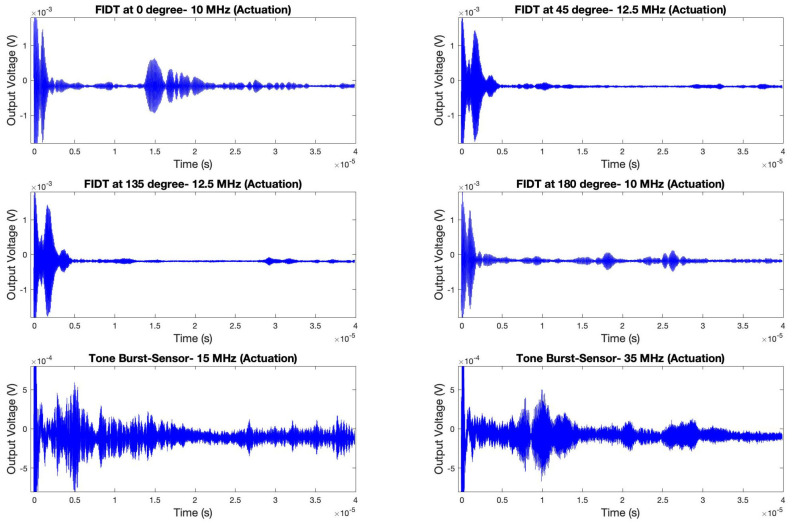
Experimental time domain signals of the different sensing electrodes at their tuned frequency.

**Figure 17 sensors-24-00644-f017:**
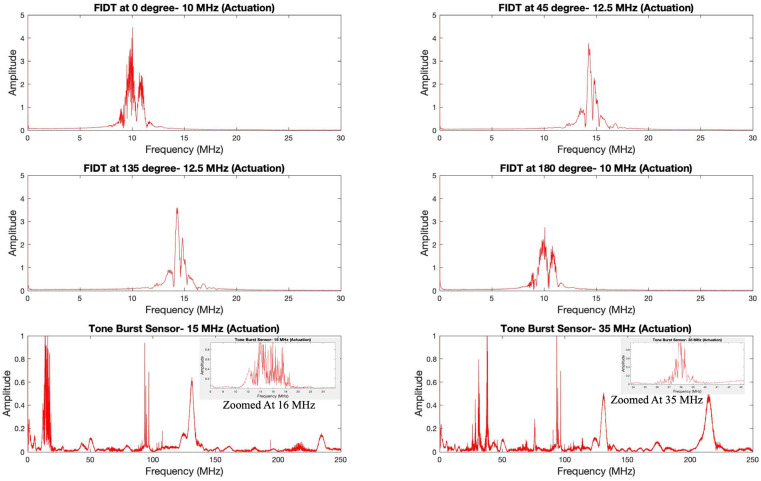
Frequency-transformed plots of the time domain signals of different sensory interdigitated electrodes at their tuned frequency.

**Figure 18 sensors-24-00644-f018:**
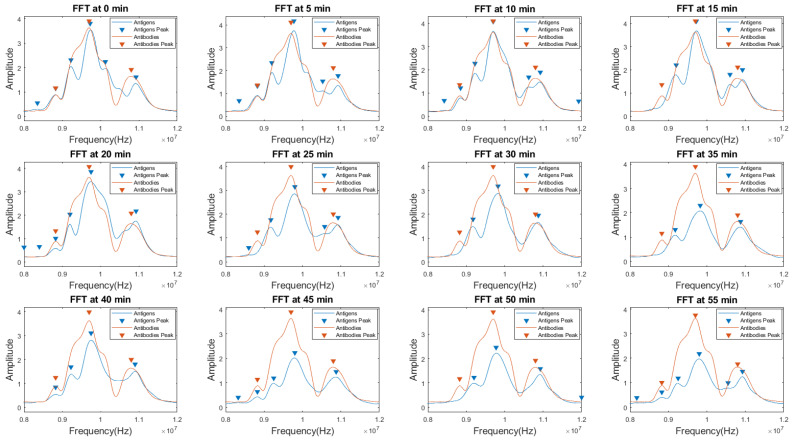
Hilbert Transformation of the signals in frequency domain of the MC-LR antigens vs. antibodies for the detection recorded over the time (0–55 min).

**Figure 19 sensors-24-00644-f019:**
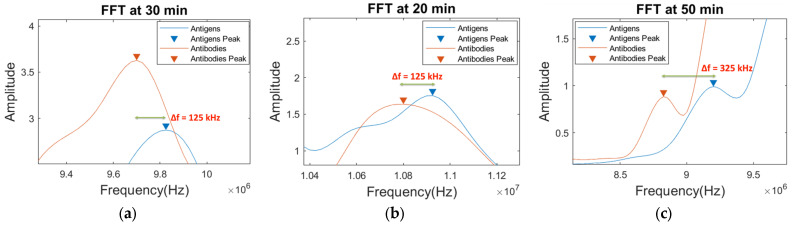
Zoomed peak frequency shifts of MC-LR antigens vs. antibodies at time interval (**a**) 30 min-near 9.75 MHz zone, (**b**) 20 min-near 10.9 MHz zone, and (**c**) 50 min-near 9 MHz zone.

## Data Availability

Dataset available on request from the authors.

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
