# Peer review of "Surface Acoustic Waves (SAW) Sensors: Tone-Burst Sensing for Lab-on-a-Chip Devices"

_sensors, 2024, doi:10.3390/s24020644_

Round 1

Reviewer 1 Report

Comments and Suggestions for Authors

Comments on the Quality of English Language

Minor editing of English language required

Author Response

Please find the attached response 

Reviewer 2 Report

Comments and Suggestions for Authors

This paper explores a challenging subject for a guided wave sensing platform that utilizes simultaneous tone burst-based excitation at multiple directions and sensing.

In my opinion, the paper provided both simulation and experimental validations on their newly proposed device. However, I am wondering how the author evaluate sensitivity, and can provide actual target analytes to show as a proof-of-concept. 

Some comments are:

1. Some descriptions are redundant or can be concised. Concise description can help the readers to catch the main story and conclusion. 

ex. Different acoustic 658 wave modes with their unique profiles were utilized to design and placement of different 659 sensing sites and sensory IDTs. ---different, unique gives no information... 

2. Despite lots of works done, there are no numbers, limit of detection, sensitivity evaluation in the text. 

Comments on the Quality of English Language

same 

Author Response

Please find the attached response 

Reviewer 3 Report

Comments and Suggestions for Authors

This paper proposed a SAW sensing device for the lab on chip systems. The design, simulation, fabrication and characterization are shown. However, the paper is not good enough for publication. The following is my main comments.

1. About the Introduction.

a. The first two paragraphs, as a background, is not very close with the theme of the paper. It is not necessary to provide all the related things in the Introduction. The contains should focus on the paper theme.

b. The state-of-art of current research is not clearly shown. Only few papers (maybe only ref. 18) are referred in the literature review part. Further reviewing work is needed.

c. The motivation is vague. I cannot clearly get the reason for proposing your work.

2.About the device design.

a. Two kinds of IDTs are used, but an explanation for it is not clearly shown. What is the difference in their roles?

b. Many wave forms are referred, e.g. coda wave, guide wave, SH wave. This brings more difficulties in understanding your paper.

c. The effect of using TB-IDT in the device is not clearly shown. If you want to say using TB-IDT is the main contribution of your paper. More qualitative researching results, e.g. theoretical analyses, simulations and then experiments are needed to prove the superiority.

3. About the figures. Many of the paper are poor in resolution, which are hard to read. Then, the photo of SAW device are shown twice. 

4.About the experiments and results.

a. What’s the relationship between Figs 16 and 17?  The frequencies are referred in the paper. Why gives the time domain results? What’s their information?

b. According to the paper, the direction affects the performances of IDTs. How to prove the superiority of TB-IDTs by locating them in the position different with other FIDTs?

c. The results from theoretical, simulated and experimental results are given in the paper. But a comparison or validation between them are not clearly given. It is necessary to glue all the parts together.  

d. The paper means to propose a sensing platform for Lab on Chip. However, only the IDT responses under ultrasonic stimulations are given. No sensing targets are added to the device. How can we evaluate its real sensing performances and verify its practicality?

e. A comparison between other published works should be done to show the superiority of TB-IDTs. 

Comments on the Quality of English Language

The writting is  too redundancy and lacked of logicality. Many long sentences are used, further making the paper hard to read.  

Author Response

Please find the attached response 

Reviewer 4 Report

Comments and Suggestions for Authors

see attachment

Author Response

Please find the attached response 

Round 2

Reviewer 2 Report

Comments and Suggestions for Authors

Thank you for your detailed response.

Introduction can be much straightforward and no need to explain every details, and it looks better now. 

I still recommend showing the numbers in the results because each application requires different frequency ranges. Based on the results with current configuration, readers can identify whether this device can be used for their needs. Authors are also encourage to discuss how to further tune these frequencies and amplitudes.

Author Response

Please find the attached response 

Reviewer 3 Report

Comments and Suggestions for Authors

I cannot get a very satisfactory response from the first round of revision. The following issues still need a consideration.

1. Is it real necessary to mention everything the in paper? In the Introduction, 1.1 and 1.2 could be replaced by two or three sentences and necessary references, due to their distance from the theme of the paper. Also, for the waves. It is true that the SAW devices work on many different parameters of waves. But, the words should be more focused on the key one with great improvement in the paper (unless everyone gets the promotion).

2. About the practical sensing experiments. I still think it is necessary to conduct some real sensing experiments (No matter “biosensing, gas sensing, chemical sensing, microfluidics, or RF sensing”) to prove the proposed device can improve the sensitivity or address the problem mentioned several times “it has been found that at times due to particular mass loading, geometry, frequency, type of waves, the central frequency shift is not significant leading to false-negatives and vice-versa”.

3. It also necessary to add the sensitivity comparison with published works, based on the exact values, to show the superiority (No matter experiment or other ways).   

Author Response

Please see the response attached
